# A Bibliometric Analysis of Research on the Role of BDNF in Depression and Treatment

**DOI:** 10.3390/biom12101464

**Published:** 2022-10-12

**Authors:** Teng He, Zifeng Wu, Xinying Zhang, Hanyu Liu, Yuanyuan Wang, Riyue Jiang, Cunming Liu, Kenji Hashimoto, Chun Yang

**Affiliations:** 1Department of Anesthesiology and Perioperative Medicine, The First Affiliated Hospital of Nanjing Medical University, Nanjing 210029, China; 2Department of Radiation Oncology, The First Affiliated Hospital of Nanjing Medical University, Nanjing 210029, China; 3Division of Clinical Neuroscience, Chiba University Center for Forensic Mental Health, Chiba 260-8670, Japan

**Keywords:** brain-derived neurotrophic factor, depression, treatment, bibliometric analysis, research trends

## Abstract

Brain-derived neurotrophic factor (BDNF), as the most widely distributed and widely studied neurotrophic factor in the mammalian brain, plays a key role in depression and the mechanisms of action for antidepressants. Currently, there is a large number of studies on the role of BDNF in the pathogenesis and therapeutic mechanism of depression. The quantity and quality of these studies, however, are unknown. To give beginners a quicker introduction to this research topic, we therefore performed a bibliometric analysis. A total of 5300 publications were included. We obtained the publications on this topic from the Web of Science database, and a variety of bibliographic elements were collected, including annual publications, authors, countries/regions, institutions, journals, and keywords. Moreover, we found that oxidative stress and neuroinflammation are the hotspots in the field in very recent years. Collectively, this study provides a comprehensive summary and analysis on the role of BDNF in depression and its treatment and offers meaningful values for beginners on this topic.

## 1. Introduction

Depression is a common type of mood disorder with the main clinical manifestation of a significant and lasting low mood, often accompanied by slow thinking, reduced volitional activity and other physical symptoms, such as loss of appetite and weight loss [1]. The characteristics of depression are an increase in negative emotions (i.e., low mood) and a decrease in positive emotions (i.e., anhedonia) [2,3]. The incidence of depression is extremely high: over 300 million people suffer from depression worldwide [4]. The harm of depression should not be ignored, and people with severe depression may even commit suicide. The World Health Organization (WHO) has published data showing that approximately 800,000 people commit suicide each year because of depression, showing that it is obviously becoming a serious global public health problem. Therefore, the research on the pathogenesis and treatment strategy of depression is a major concern in current society.

Brain-derived neurotrophic factor (BDNF), one of the most widely distributed and extensively studied neurotrophic factors in the mammalian brain, is a highly conserved protein synthesized from 247 amino acids [5]. The synthesis and maturation of BDNF undergo a multistage process, including being synthesized and folded into preproBDNF in the endoplasmic reticulum, rapidly cleaved into the isoform proBDNF in the Golgi apparatus, and further cleaved to reach the mature isoform (mBDNF) [6,7,8]. BDNF plays an important role in neuronal maturation, synapse formation and synaptic plasticity in the brain [9]. Studies have shown that BDNF regulates different cellular processes involved in the development and maintenance of normal brain function mainly by binding and activating the tropomyosin receptor kinase B (TrkB) [8].

In addition to depression, deficits in BDNF signaling have been reported to contribute to several neuropsychiatric diseases, such as Huntington’s disease, Alzheimer’s disease (AD), schizophrenia and anxiety disorders [10,11,12,13,14,15]. It is well acknowledged that there is a strong linkage between BDNF and depression [16,17,18,19,20,21]. It has been shown that the drugs increasing the plasma concentration of BNDF can improve depressive symptoms [22,23]. Within the central nervous system, a reduction in the expression levels of BDNF and its receptor TrkB has been reported in the hippocampus and prefrontal cortex of post-mortem brain tissues of suicide victims [24,25]. Furthermore, there is a report showing altered expression of BDNF and its precursor proBDNF in the postmortem brain and liver from depressed patients [26]. In contrast, the involvement of BDNF in the efficacy of antidepressant treatment has mainly been demonstrated in rodent models, and that clinically prescribed antidepressants exert improving effects to increase TrkB autophosphorylation and signaling within hours in selected brain regions [27,28]. There are reports showing decreased blood levels of BDNF in patients with depression [29,30,31]. To sum up, BDNF serves as a transducer, acting as the link between the antidepressant drug and the neuroplastic changes that result in the improvement of the depressive symptoms [21].

Bibliometric analysis firstly emerged in the early 20th century, and formed an independent discipline in 1969, which is widely used in literature analysis [32]. Bibliometric analysis provides a quantitative approach to the study of published literature in a specific field [33]. With the help of computer technology, it displays the results of a visual analysis of the literature through simple and clear maps for the purpose of describing and analyzing the progress of a certain research topic [34]. In addition, bibliometric analysis can evaluate academic productivity, summarize academic frontiers and hotspots of research, and obtain detailed information about countries, institutions, authors, journals, keywords, etc. Interestingly, other methods, such as experimental studies, mate analysis, and traditional reviews, both fail to achieve the same depth of analysis [35].

Recently, although numerous reviews provide a relatively comprehensive description regarding the role of BDNF in depression [8,20,36], there is no evidence of high quality on the bibliometric analysis of this topic. In this study, 5300 publications on the role of BDNF in depression and treatment were evaluated by bibliometrics as of 2 August 2022, so as to further clarify the research hotspots and trends in the association between depression and BDNF.

## 2. Materials and Methods

### 2.1. Data Source and Collection

Web of Science Core Collection was selected as the data source in this study. In the meantime, the “citation index” was set as SCI-EXPANDED to ensure the comprehensiveness and accuracy of the retrieved data. It must be pointed out that as a high-quality digital literature resource database, Web of Science has been accepted by the majority of researchers and has been considered to be the most suitable database for bibliometric analysis [37]. Therefore, we selected Web of Science as the data source. We adopted “(TS = (Depression)) AND TS = (BDNF)” as the retrieval strategy and restricted the literature type to “article OR review”. However, there were no language restrictions during the process, and we have not set a time limit, only a deadline of the retrieval data (2 August 2022). All retrieved records were downloaded in “plain text” format and relevant information was extracted for further analysis (Figure 1).

### 2.2. Data Extraction

Literature selection and data extraction were carried out by two independent researchers to ensure the reliability of the results. We extracted and analyzed the elements from the acquired literature including the number of annual publications, authors, countries/regions, institutions, journals, the number of citations, keywords and so on. In order to objectively evaluate the scientific output and academic achievement of authors, countries/regions and institutions in the field of depression and BDNF, the H-indexes corresponding to these elements were collected. Furthermore, the Journal Impact Factor and the Journal Citation Reports of 2021 for the top 10 journals in terms of number of articles were also collected in this study.

### 2.3. Data Analysis and Visualization

VOSviewer 1.6.17 (Nees Jan van Eck and Ludo Waltman, Leiden University, The Netherlands) and CiteSpace 5.8.R3 (Chaomei Chen, Drexel University, Philadelphia, PA, USA) can give full play to their respective advantages in the drawing of knowledge mapping, so we used VOSviewer and CiteSpace for knowledge mapping. VOSviewer is a probabilistic-based data standardization approach which can provide a variety of visualizations including Network Visualization, Overly Visualization, and Density Visualization in terms of co-authors, co-countries and the co-occurrence of keyworks [38]. CiteSpace is a data standardization approach based on set theory, which allows for a Timezone and Timeline view within a time slice to understand the development process and trends of this field [39]. It should be mentioned that VOSviewer and CiteSpace, the most commonly used bibliometric tools, are available [40]. Microsoft Excel 2019 (Microsoft Corporation, Redmond, WA, USA) was used for data processing apart from VOSviewer and CiteSpace.

## 3. Results

### 3.1. Annual Publications and Trends

In this study, a total of 5300 publications on the role of BDNF in depression and treatment were collected from the Web of Science core collection (WoSCC), which were published in 822 journals between 1999 and 2022. It is noteworthy that “Article” was the predominant type in the 5300 publications, accounting for 85.30% (4521/5300). A total of 13 languages were used in these publications, among which English was the most predominant language, accounting for 98.89% (5241/5300).

As shown in Figure 2, research trends in the topic of the association between depression and BDNF showed a year-on-year over the years of retrieval. It should be noted that the reason underlying the low number of publications in 2022 is the search deadline for this study is 2 August 2022, rather than the whole year. The year-on-year increase in the number of publications reflects, to some extent, the growing scientific attention on depression and BDNF.

### 3.2. Contributions of Authors

During the set search period, a total of 24,890 authors were involved in the study of the topic. According to Price Law [41], the authors who have published six or more in the topic were core authors. The top 15 authors in terms of publications are shown in Table 1. An analysis of the co-authorship network provides information on the representative scholars and core research strengths of the research field. VOSviewer was adopted to visualize the co-authorship network for authors with more than 10 publications. Network visualization is able to show each individual cluster. In this context, it is possible to identify research cliques by examining author collaborations and to find out the similarities and differences in scholars on research topics by analyzing the author coupling network. Some of the 145 authors in our network were not connected to each other, and the largest set of connected items consisted of 95 authors. For this, we only showed the largest set of authors instead of all (Figure 3). These authors were grouped into 14 clusters. Among them, the two clusters with the highest link strength were centered on Professor Kenji Hashimoto (Chiba University, Chiba, Japan) and Professor Joao Quevedo (University of Southern Santa Catarina, Tubarão, Brazil), respectively. It was of importance to emphasize that Professor Ronald S. Duman (Yale University, New Haven, CT, USA) had the highest average citations per paper reaching 277.27, which was much higher in the number of citations.

### 3.3. Contributions of Countries or Regions

A total of 86 countries/regions were involved in the study. Firstly, a visual analysis of the countries with 18 or more publications was conducted by VOSviewer, and the results are shown in Figure 4. The distribution of publications in this field was very uneven in terms of countries, and the top effect was very significant. Most of the publications were from a few countries. Then, we further analyzed the high-productivity countries in this field, and the top 15 countries/regions in terms of publications are shown in Table 2 and Appendix A. It is noteworthy that the 95% confidence interval for average citation/publication was [31.1786, 47.3294]. It is quite clear that the United Sates was the most productive and influential country with a total of 1296 relevant papers, and the H index was 147. In addition, the United States also had the highest average citations per paper reaching 67.73.

### 3.4. Contributions of Institutions

During this period, a total of 4489 institutions were involved in the topic. The top 10 institutions are shown in Figure 5. As an aside, the 95% confidence interval for average citation/publication was [31.6470, 93.9790]. It is worth noting that these institutions are from different countries. Federal University of Rio Grande do Sul (Rio Grande do Sul, Brazil) was considered the most productively institution in the field with 110 publications, whereas Yale University (USA) was considered the most influential institution with an H index of 57. On average, documents from Yale University have been cited 175.42 times, which was much higher in the number of citations. It should be mentioned that Professor Ronald S. Duman from Yale University made an outstanding contribution in this respect.

### 3.5. Analysis of Journals

Until the deadline (2 August 2022), a total of 882 journals have published 5300 pieces of literature related to the association between depression and BDNF. The top 10 most productive journals have published 1041 related documents, accounting for 19.64% of all publications (Table 3). Obviously, *Behavioural Brain Research* has been the most productive journal with 155 documents on the subject. On average, documents from *International Journal of Neuropsychopharmacology* were cited 56.78 times, which was the highest among these 10 journals. Most of the top 10 productive journals were classified as Q2 according to the JCR 2021 standards. Furthermore, we spotted that the journal with the highest impact factor was *Journal of Affective Disorders*, which had an impact factor of 6.533.

### 3.6. Analysis of Keywords

Keywords condense the core and essence of a paper, and the research hotspots in a certain scientific field can be revealed by keyword co-occurrence analysis. VOSviewer was used to create a keyword co-occurrence network view of 5300 documents, and 267 keywords with a frequency more than or equal to 40 were selected for visualization (Figure 6A). In Figure 6A, the larger the circular node, the more times the keywords, and the more representative the hotspots are. The line of nodes represents the strength of association, and the thicker the line is, the more times they appear together in the same document. Node colors represent different clusters, namely research topics. Additionally, in this figure, it was found that the identified keywords were divided into five clusters and it was dominated by three areas: green, red and blue. Keywords clustered in the green area mainly described topics related to animal experiments; in the red area, clustered keywords were related to clinical classification and characteristics of depression; in the blue area, clustered keywords were related to synaptic plasticity.

In order to reveal the research progress on the association between depression and BDNF in the time dimension, the Timezone graph supplied by CiteSpace was used to display the map of the keyword co-occurrence network (Figure 6B). The gradual evolution of the association between depression and BDNF from the field of synaptic plasticity to the field of oxidative stress and neuroinflammation was revealed.

Burst keyword show the phenomenon that any keyword emerges frequently in a particular time of period. Therefore, not only the evolution of research hotspots with time can be found, but also the research trends in recent years can be studied and suggestions may be made for that of the future. The top 25 burst keywords were evaluated in this study (Appendix A). In this regard, we drew a conclusion that future research in the field would be closely linked to neuroinflammation.

## 4. Discussion

Bibliometrics is an effective measurement method used to describe and analyze the dynamic development of a particular research field. With the help of modern computer technology, it can visually analyze documents in a clear and concise knowledge map [34]. Through the knowledge map provided by bibliometrics, researchers can comprehensively understand the basic situation, current hotspots, and future trends of a research field, which breaks through the limitations of time and space [42]. Bibliometrics therefore plays a very important role in helping beginners to enter a certain field quickly and read documents more efficiently.

The association between depression and BDNF has attracted increasing academic attention over the years. In addition, there have been a large number of documents on this very detailed elaboration at present. In this study, we have already collected the documents on the association between depression and BDNF which have published in the Web of Science database. In this regard, we also have analyzed the basic information about these documents, including changes in annual publications, the most productive and influential authors, countries/regions, institutions, journals and keywords. For this, the evolution process of research hotspots on the role of BDNF in depression was evaluated.

The development process of the research field of the association between depression and BDNF can be reflected by the change of annual output. It is quite clear that the annual output showed an increasing trend year by year. We speculate that this has a lot to do with the current interest in depression and the strong link between depression and BDNF. It is therefore likely that research on the association between depression and BDNF may continue to be emphasized in the future.

Professor Joao Quevedo from University of Extremo Catarinense in Brazil and Professor Kenji Hashimoto from Chiba University in Japan are the top two authors in terms of the number of publications. Interestingly, the two groups with the highest link strength were centered on Professor Kenji Hashimoto and Professor Joao Quevedo, respectively. In addition, Professor Kenji Hashimoto has a higher H index, which goes some way to demonstrate that Professor Kenji Hashimoto has a very high level of influence in the field. It is noteworthy that Professor Ronald S. Duman from Yale University in the United States has published literature with extra high average frequency of citations, indicating that the quality of the documents from him is very high and his documents are worth studying. For example, Shirayama’s article showing a long-lasting antidepressant-like effect of BDNF in learned helplessness model rats is highly cited [43].

The United States has the highest number of publications, H index and average frequency of citations, which fully demonstrates that the United States is the most productive and influential country in the field. This phenomenon indicates that the United States has the highest quality literature in this field and has made the greatest contribution. This has a lot to do with the fact that the United States has been a leader in medical research over the last few decades. It is likely that the cooperation between different countries is very extensive. This phenomenon illustrates that depression has attracted attention worldwide, and that it is necessary to strengthen international cooperation.

Federal University of Rio Grande do Sul in Brazil published the most documents of all institutions, indicating that Federal University of Rio Grande do Sul is the most productive institution on the topic. However, Yale University in the United States has the highest H index and average frequency of citation, which illustrates that Yale University is the most influential institution and the quality of the literature published by Yale University are widely recognized by peers.

In terms of journals, *International Journal of Neuropsychopharmacology* (IF: 5.678, Q1), *Neuroscience* (IF: 3.708, Q3), *Neuropharmacology* (IF: 5.273, Q2) and *Progress in Neuro-Psychopharmacology & Biological Psychiatry* (IF: 5.273, Q2) have a relatively high average frequency of citations compared to other journals. On average, documents published in these journals were cited more than 50 times. Generally speaking, highly cited literature indicates that it has received a lot of attention in the research field, which is well worth learning for beginners. Therefore, beginners should pay more attention to the literature published in these journals.

The core and essence of a document can be reflected by keywords. Keywords co-occurrence analysis can reveal the research hotspots in a scientific field. The development tendency of research hotspots can be presented more intuitively by keyword co-occurrence network diagram of Timezone [39]. On the other hand, burst keywords can make suggestions for future research on the basis of the evolution of research hotspots. Through the analysis of keywords, we speculated that oxidative stress and neuroinflammation would attract increasing attention in the future. In order to provide the information about the most influential papers in this field, we have provided the co-citation analysis among the citied references in the Appendix A. It has been well recognized that papers with the highest degree of centrality within each cluster would possess the highest impact in the field. There is no doubt that these papers would be of great value to readers.

Depression is a common type of mood disorder, and more than 300 million people worldwide suffer from depression [4]. In addition, approximately 800,000 people commit suicide per year as a result of depression. As a consequence, the research on depression never stops. As the most widely distributed and widely studied neurotrophic factor in the mammalian brain, BDNF is also highly correlated with depression, and is therefore often investigated in the studies of depression. However, the current studies on the involvement of BDNF in the efficacy of antidepressant treatment are mainly focused on rodent models, and the relevant clinical studies are relatively rare. It is worth noting that behavioral tests used in rodent models of depression may not accurately predict antidepressant effects in depressed individuals, as the full complexity of human depression is not fully properly mimicked in rodents [44,45,46,47]. In other words, due to the complexity of human psychiatric disorders, data from rodents may not necessarily applicable for humans. Therefore, in the future, it is essential to strengthen the clinical research in related aspects.

## 5. Limitation

Due to the existence of some objective factors, this study also has a number of limitations. Firstly, Web of Science Core Collection was selected as the data source in this study, and the “citation index” was set as SCI-EXPANDED which starts retrieving data in 1999. This situation inevitably raises the issue of incomplete data analysis. However, it does not mean that those not included documents are not important. Moreover, due to the continuous update of data in the WoSCC database, the results of bibliometric analysis often lag behind the actual research progress. The most important point is that high-quality bibliometric analysis requires researchers to have a comprehensive and in-depth knowledge of the field under study.

## 6. Conclusions

In this study, 5300 published documents were collected using bibliometrics to clarify the current status and future trends of research on the role of BDNF in depression. There is no doubt that researchers have been paying attention to depression. BDNF is closely related to depression, so the research on depression and BDNF has been a research hotspot in recent years. At the same time, we found that the focus of research on the association between depression and BDNF has gradually shifted from the field of “synaptic plasticity” to “oxidative stress and neuroinflammation”. Therefore, we speculate that future research will focus more on oxidative stress and neuroinflammation. In conclusion, our study will provide a valuable reference for future studies on the association between depression and BDNF.

## Figures and Tables

**Figure 1 biomolecules-12-01464-f001:**
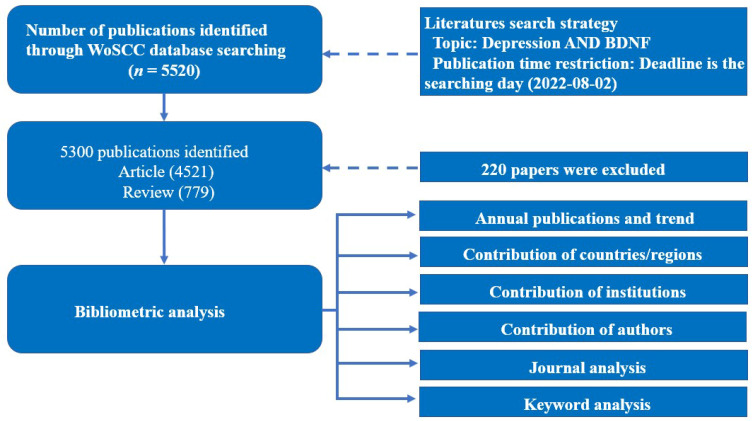
Flow diagram of literature identification.

**Figure 2 biomolecules-12-01464-f002:**
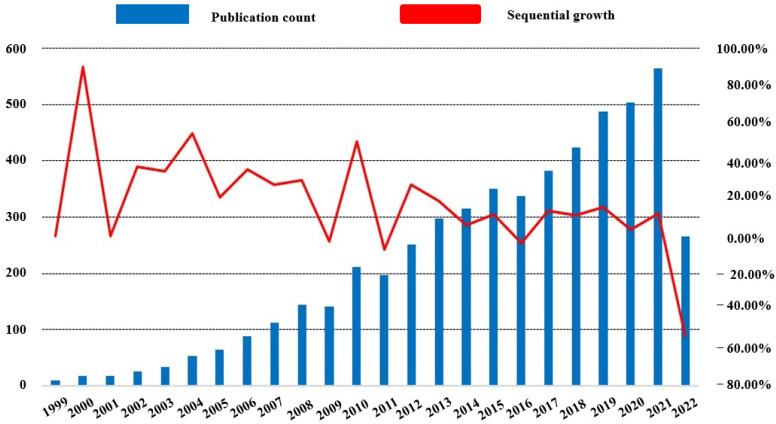
Annual publications and trends. Global trends and sequential growth rate of annual publications related to Depression and BDNF research.

**Figure 3 biomolecules-12-01464-f003:**
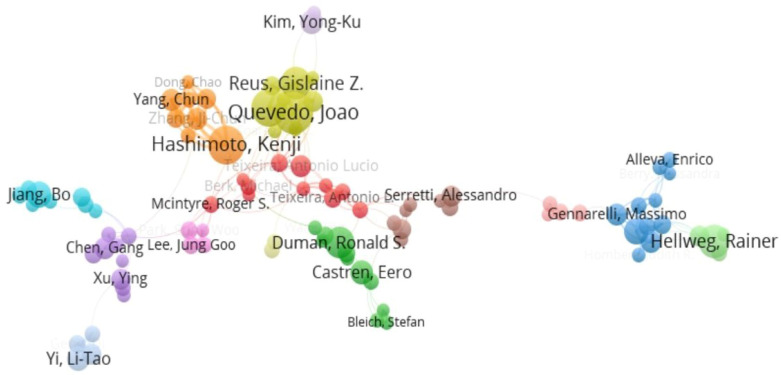
Contributions of authors. Mapping of the co-authorship analysis among the authors who published at least 10 papers on the association between depression and BDNF according to VOSviewer. In the figure, the node sizes represent the volume of published literature, the larger the node, the greater the volume of published literature; the lines represent the association strength, the thicker the line, the stronger the connection between the two authors; the node colors represent different clusters.

**Figure 4 biomolecules-12-01464-f004:**
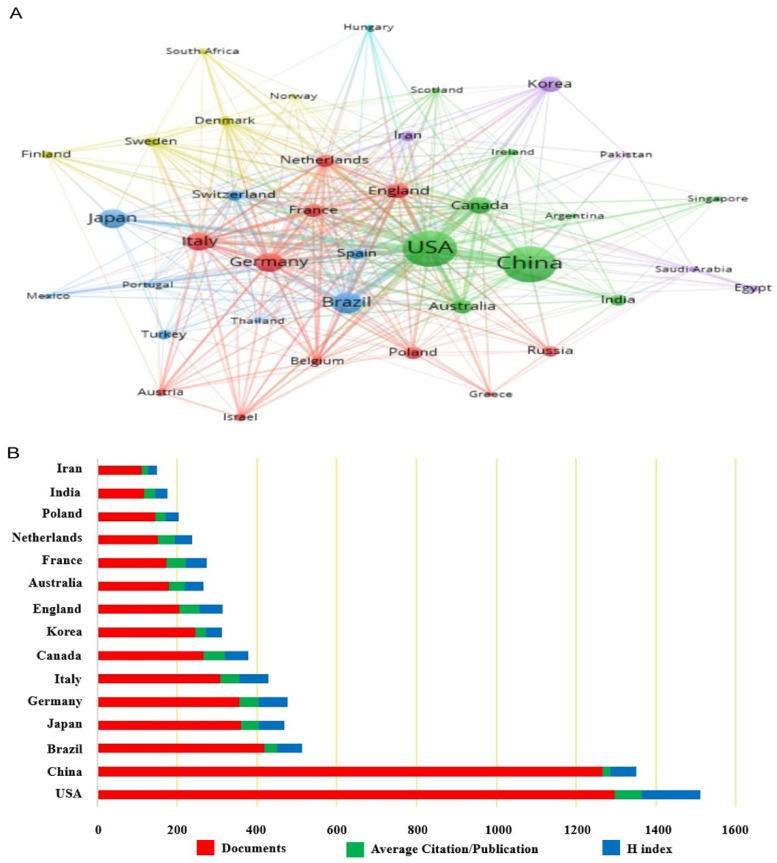
Contributions of countries or regions. (**A**) Mapping of the co-authorship analysis among the countries which published at least 18 papers on the association between depression and BDNF according to VOS viewer. Node sizes represent the volume of published literature. Node lines represent association strength. Node colors represent different clusters. (**B**) The total number of publications, average citation per item, and H index of the top 15 countries.

**Figure 5 biomolecules-12-01464-f005:**
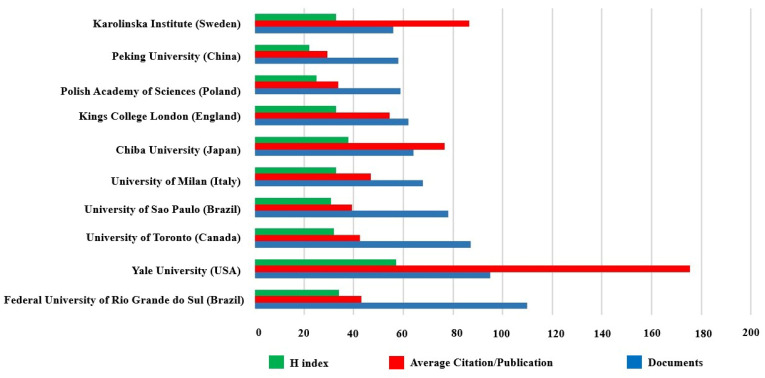
Contributions of institutions. The total number of publications, average citation per item, and H index of the top 10 institutions.

**Figure 6 biomolecules-12-01464-f006:**
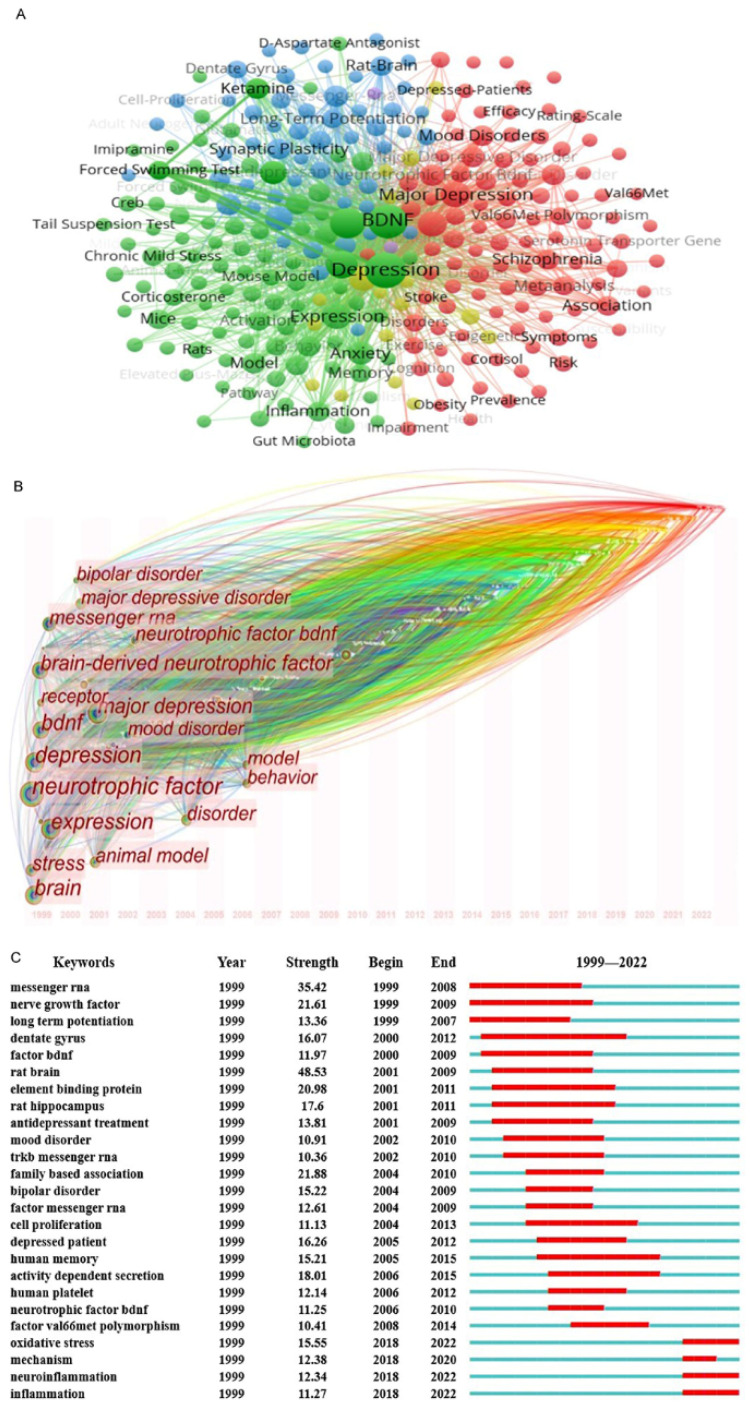
Keyword analysis. (**A**) Mapping of the timezone view on the association between depression and BDNF according to CiteSpace. (**B**) Mapping of the co-occurrence analysis among the keywords which occurred at least 40 times on the association between depression and BDNF according to VOSviewer. Node lines represent association strength. Node colors represent different clusters. (**C**) Top 25 keywords with the strongest citation bursts on the association between depression and BDNF.

**Table 1 biomolecules-12-01464-t001:** Contributions of authors. The top 15 authors for the association between depression and BDNF research.

Rank	Author	Documents	Citations	Average Citations/Publications	H Index
1	Quevedo, J	50	2337	46.74	31
2	Hashimoto, K	45	2584	57.42	35
3	Kapczinski, F	43	2983	69.37	33
4	Hellweg, R	37	1322	35.73	27
5	Reus, GZ	34	1861	54.74	26
6	Duman, RS	33	9150	277.27	33
7	Yoshimura, R	27	890	32.96	17
8	Calabrese, F	25	1161	46.44	18
9	Jiang, B	25	600	24.00	16
10	Nakamura, J	24	884	36.83	17
11	Kim, JM	24	822	34.25	16
12	Kim, SW	24	822	34.25	17
13	Castren, E	23	3009	130.83	23
14	Tsai, SJ	23	742	32.26	21
15	Hori, H	22	858	39.00	17

**Table 2 biomolecules-12-01464-t002:** Contributions of countries or regions. The top 15 countries for the association between depression and BDNF research.

Rank	Country	Documents	Citations	Average Citations/Publications	H Index
1	USA	1296	87,774	67.73	147
2	China	1266	25,722	20.32	63
3	Brazil	418	13,777	32.96	61
4	Japan	361	15,815	43.81	64
5	Germany	357	16,722	46.84	73
6	Italy	307	15,335	49.95	72
7	Canada	266	14,494	54.49	57
8	Korea	246	6188	25.15	41
9	England	205	10,567	51.55	58
10	Australia	180	7156	39.76	47
11	France	174	8171	46.96	52
12	Netherlands	151	6301	41.73	45
13	Poland	145	3701	25.52	33
14	India	118	3197	27.09	30
15	Iran	112	1674	14.95	23

**Table 3 biomolecules-12-01464-t003:** Journal analysis. The top 10 journals for the association between depression and BDNF research.

Rank	Journal	Publication Count	Impact Factor™ (2021)	Journal Citation Reports™ (2021)	Total Citations	Average Citations
1	Behavioural Brain Research	155	3.352	Q2	4681	30.20
2	Journal of Affective Disorders	131	6.533	Q1	4136	31.57
3	Progress in Neuro-Psychopharmacology & Biological Psychiatry	130	5.201	Q2	6576	50.58
4	Neuroscience Letters	112	3.197	Q3	3485	31.12
5	PLoS One	99	3.752	Q2	4308	43.52
6	Journal of Psychiatric Research	86	5.250	Q2	3928	45.67
7	Neuroscience	85	3.708	Q3	4671	54.95
8	Neuropharmacology	84	5.273	Q2	4402	52.40
9	Psychopharmacology	82	4.415	Q2	3108	37.90
10	International Journal of Neuropsychopharmacology	77	5.678	Q1	4372	56.78

## Data Availability

Not applicable.

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
