# Peer review of "A Bibliometric Analysis of Research on the Role of BDNF in Depression and Treatment"

_biomolecules, 2022, doi:10.3390/biom12101464_

Round 1

Reviewer 1 Report

The presented article is interesting and includes a proper bibliometric analysis of the research on the role of BDNF in depression and treatment. The authors analyzed a large amount of bibliographic data and used a nice and proper visualization for the results of their bibliometric analysis. Some language mistake and spell mirrors require to be improved.

Author Response

Thank you for your suggestive comments. We corrected.

Reviewer 2 Report

I will stay with the issues I find related to methodology and the scope of the research.   Firstly, please add the PRISMA diagram to explain how data has been crawled and filtered. Secondly, I also re-run the search query in WoS, but the total of 5518 records from the data 1900-01-01 till 2022-08-02 was recorded. It means the author is ignoring the starting date of the search query, which is important to mention.   Thirdly, the search query is not correctly stated in the method section, which is incorrect as per the WoS search facility. Moreover, in my understanding, the review articles should also be excluded from the primary data set for the study, as suggested by several existing bibliometric researchers.   To my understanding, there is no significant difference between sections 3 and 4 as both of these sections are just explaining what is happening. But no insight into the future implications has been discussed. See Section 5 of this reference paper Mora L, Deakin M, Reid A (2018) Combining co-citation clustering and text-based analysis to reveal the main development paths of smart cities. Technol Forecast Soc Chang. https://doi.org/10.1016/j.techfore.2018.07.019   I strongly recommend you perform thematic cluster analysis through CiteSpace to make the research more insightful, on the present moment, it's only about authors, journals, keywords, countries, and institutions. It is not much insightful except descriptive profiling, which is not acceptable to be published without any theoretical and practical leads and predictions.    I suggest you eliminate a few figures and put them into the appendix (supplementary document) with higher resolution, as most of them are not even readable.   As a reviewer, I would suggest adding BURST analysis for articles co-citation analysis, and exploring further the burst keywords in keywords co-occurrence analysis to discuss what kind of content can be predicated related to each keyword and article which are listed in the burst analysis. For a reference, please look at this reference Seyedghorban Z, Matanda MJ, LaPlaca P (2016) Advancing theory and knowledge in the business-to-business branding literature. J Bus Res 69:2664–2677. https://doi.org/10.1016/j.jbusres.2015.11.002   In my understanding, this manuscript needs serious changes to be addressed before it can be published. Looking forward to reading the revised version.

Reviewer 3 Report

I would like to congratulated authors for the theme and the study. The study made excellent impression. i do not have additional comments.

Author Response

Thank you for your efforts and comments. 

Reviewer 4 Report

In their work, He et al. performed a bibliometric analysis of publications related to the role of BDNF in depression. Using Web of Science Collection, the authors identified 5,300 publications on that topic. Utilizing corresponding metadata and network analysis, they assessed the contribution of various authors, countries and institutions, and analyzed top journals and keywords associated with the topic. In particular, based on the analysis of keywords frequency with time, they revealed the shift of research interest from synaptic plasticity to oxidative stress and neuroinflammation.

While this work provides an interesting metadata related to the papers in the field, it should better justify the importance of the topic (Major point 1) and include network analysis of the publications pointing to the most influential works in the area (Major point 2) to become a valuable resource for the audience.

Major points:

1)     Authors claim that BDNF plays a key role in depression. To justify this claim and provide a good idea on the role of BDNF in depression research and its dynamics with time, it will be valuable to complement the trend analysis with the data on proportion of papers related to depression where BDNF is mentioned. This can be done by looking at all publications from Web of Science on depression (“TS = Depression”) and quantifying those of them where ‘BDNF’ is either a keyword or is included in the title or abstract. If the proportion of such papers is substantial and increases with time, it will be a good confirmation of the authors’ claim and justify the importance of their work.

2)     Although the work provides various perspectives, including data on the authors, institutions and journals associated with the publications on depression and BDNF, it doesn’t provide the information about the most influential (core) papers in the field. Using co-citation analysis, network of individual publications can be created. Papers with the highest degree of centrality within each cluster would be the papers with the highest impact on the field. The list of such core papers together with their brief description would be of great value for the readers of this manuscript.

Minor points:

1)     Figure 4 (“Contributions of institutions”) is called “Figure 5” instead. In addition, error bars (standard errors or confidence intervals) are not shown for ‘average citation/publication’ values on this figure. They would allow to assess the dispersion of this metric across papers and, therefore, would add substantial value to the plots.

2)     On fig. 3B, stacked barplot looks confusing, since the presented metrics (number of documents, average number of citations per publication and H-index) have different scale. 3 separate barplots would be more appropriate in this case. Error bars should also be included for ‘average citation/publication’ values.

3)     Current version of the manuscript has many typos, they should be corrected (e.g., “These are currently large numbers of studies” in Abstract and “BDNF serves as a transducer linking the antidepressants and the alterations in neuroplasticity, finally leads to the clinical improvement of depressive symptoms” in Introduction).

Round 2

Reviewer 2 Report

I tried to give very sincere and valuable reviews which actually needed significant changes, which were noted as really value to be added.

The way the authors tired to add reviews is not satisfactory. 

For further references, please have a look at the reviews I made in the previous round.

The present changes, as it wasn't minor change. 

I recommend it be rejected this time.

Author Response

Thank you for your suggestive comments. We have revised our manuscript as your comments.